# An Efficient CRISPR/Cas9 Genome Editing System for a *Ganoderma lucidum* Cultivated Strain by Ribonucleoprotein Method

**DOI:** 10.3390/jof9121170

**Published:** 2023-12-05

**Authors:** Yi Tan, Xianglin Yu, Zhigang Zhang, Jialin Tian, Na Feng, Chuanhong Tang, Gen Zou, Jingsong Zhang

**Affiliations:** 1National Engineering Research Center of Edible Fungi, Institute of Edible Fungi, Shanghai Academy of Agricultural Sciences, Shanghai 201403, China; judy_1989317@126.com (Y.T.); fengna006@163.com (N.F.); 2College of Food Sciences & Technology, Shanghai Ocean University, Shanghai 201306, China; shiny_yuu@163.com (X.Y.); jialint201306@163.com (J.T.); 3College of Life Sciences, Shanghai Normal University, Shanghai 200234, China; 18734283267@163.com

**Keywords:** *Ganoderma lucidum*, CRISPR/Cas9, genome editing, ribonucleoproteins (RNPs), the orotidine 5′-monophosphate decarboxylase gene (*ura3*)

## Abstract

The CRISPR/Cas9 system has become a popular approach to genome editing. Compared with the plasmid-dependent CRISPR system, the ribonucleoprotein (RNP) complex formed by the in vitro assembly of Cas9 and single-guide RNA (sgRNA) has many advantages. However, only a few examples have been reported and the editing efficiency has been relatively low. In this study, we developed and optimized an RNP-mediated CRISPR/Cas9 genome editing system for the monokaryotic strain L1 from the *Ganoderma lucidum* cultivar ‘Hunong No. 1’. On selective media containing 5-fluoroorotic acid (5-FOA), the targeting efficiency of the genomic editing reached 100%. The editing efficiency of the orotidine 5′-monophosphate decarboxylase gene (*ura3*) was greater than 35 mutants/10^7^ protoplasts, surpassing the previously reported *G. lucidum* CRISPR systems. Through insertion or substitution, 35 mutants introduced new sequences of 10–569 bp near the cleavage site of *ura3* in the L1 genome, and the introduced sequences of 22 mutants (62.9%) were derived from the L1 genome itself. Among the 90 mutants, 85 mutants (94.4%) repaired DNA double-strand breaks (DSBs) through non-homologous end joining (NHEJ), and five mutants (5.6%) through microhomology-mediated end joining (MMEJ). This study revealed the repair characteristics of DSBs induced by RNA-programmed nuclease Cas9. Moreover, the *G. lucidum* genes *cyp512a3* and *cyp5359n1* have been edited using this system. This study is of significant importance for the targeted breeding and synthetic metabolic regulation of *G. lucidum*.

## 1. Introduction

*Ganoderma lucidum* is one of the best-known medicinal macrofungi in the world, and is included in the American Herbal Pharmacopoeia and Terapeutic Compendium [1]. The *G. lucidum* has been found to have various therapeutic activities such as anti-tumor, antiviral, antihypertensive, and immune regulation, through modern pharmacological research [2]. *G. lucidum* products are consumed worldwide as nutraceuticals and daily supplements, with annual sales of more than 2.5 billion dollars [3]. *G. lucidum* contains a large quantity of bioactive substances, of which more than 400 compounds have been identified. Triterpenoids and polysaccharides are the two major active substances of *G. lucidum* [4]. The study of the *G. lucidum* gene function is the basis for exploring its various life activities, and the lack of mature genetic tools for gene function research has seriously impeded the development of applied sciences such as directed breeding and anabolic regulation in *G. lucidum* [5].

The CRISPR/Cas9 genome editing system is an important tool for gene function research. The Cas9 protein and single-guide RNA (sgRNA) induce DNA double-strand breaks (DSBs) at the targeted sequence of a genome, allowing for efficient and precise insertion, knockout, and site-directed mutation operations using different donor DNAs [6]. As the third-generation genome editing system, CRISPR/Cas9 has been widely applied in animals, plants, and microorganisms [7,8,9]. Researchers have constructed CRISPR/Cas9 systems in different strains of *G. lucidum*: strain CGMCC 5.26 is widely used as *G. lucidum* herbs in China, and 260125 is its monokaryotic strain [4]. By using the Cas9 expression plasmid and in vitro-transcribed sgRNA, a CRISPR/Cas9 genome editing system was successfully constructed in the strain 260125, with an editing efficiency of 0.2–18 mutants/10^7^ protoplasts [5,10]. *G. lucidum* “Hunong No. 1” is widely cultivated, accounting for 70% of the high-spore powder varieties of China, and L1 is its monokaryotic strain [11]. By using the Cas9 expression plasmid and in vitro-transcribed sgRNA, a CRISPR/Cas9 genome editing system was successfully constructed in the strain L1, with an editing efficiency of 4 mutants/10^7^ protoplasts [11]. The addition of the surfactant Triton X-100 (0.006%) increased the editing efficiency to greater than 18 mutants/10^7^ protoplasts, indicating that this concentration of Triton X-100 effectively increased cell membrane permeability, allowing more sgRNAs to enter protoplasts and thereby improving transformation efficiency [11]. To achieve the transcription of sgRNA in *G. lucidum*, Wang et al. anchored the *u6* gene and its promoter of *G. lucidum* strain CGMCC 5.616 based on the conserved sequences of the *u6* gene in different species [12], and added HDV ribozyme at the 3′ end of sgRNA, which accomplished the expression of *cas9* and transcription of sgRNA on one plasmid, with an editing efficiency of 5.3 mutants/10^7^ protoplasts. The *G. lucidum* dikaryotic strain GL3315 is from the Sorak Mountains located in the northwestern part of Korea. Eom et al. successfully constructed a CRISPR/Cas9 genome editing system for the monokaryotic strain of GL3315 using ribonucleoproteins (RNPs) assembled by the Cas9 protein and sgRNA [13], with the editing efficiencies of 0, 0.33, and 22 mutants/10^7^ protoplasts for different target sites, and its transformation system was supplemented with 0.90% of Triton X-100.

The CRISPR/Cas9 systems of *G. lucidum* are constructed using the orotidine 5′-monophosphate decarboxylase gene (*ura3*) as the target gene [5,10,11,12,13]. The *ura3* gene of *G. lucidum* is involved in catalyzing a key reaction in the synthesis of uracil and can convert 5-fluoroorotic acid (5-FOA) into the toxic compound 5-fluorouracil (a suicide inhibitor), causing cell death [14]. Based on these characteristics, *ura3* can be used as a dual selection marker: when *ura3* is disrupted, the strain becomes uracil auxotrophic and can grow normally on media containing 5-FOA; when the *ura3* gene is restored, the strain is able to synthesize uracil and can grow normally on media without uracil.

The reported CRISPR/Cas9 systems of *G. lucidum* mostly utilize the Polyethylene Glycol (PEG)-mediated transformation method to deliver the *cas9* expression plasmid into the *G. lucidum* protoplast [5,10,11,12]. The exogenous genes such as *cas9* and *sdhB* are typically randomly integrated into the *G. lucidum* genome with multiple copies after PEG-mediated transformation [15], which may bring certain risks. Firstly, the introduction of exogenous genes may impact the genomic structure of *G. lucidum*, and their continuous expression may lead to unexpected phenotypes or even toxicity in the host cells [16,17]. Secondly, under the action of endogenous nucleases of *G. lucidum*, the plasmids may be degraded into small DNA fragments and inserted into the genome, resulting in unpredictable changes [18]. An in vitro-expressed Cas9 protein and transcribed sgRNA can be assembled into RNP complexes. The RNP-mediated CRISPR/Cas9 system has advantages such as not introducing exogenous sequences, not relying on codon optimization of *cas9* based on whole-genome sequencing, and not depending on the endogenous promoter of sgRNA. The first mushroom-forming fungus for which RNPs were developed was *Schizophyllum commune* [19], and other filamentous fungi have also been constructed with the RNP system, such as *Trichoderma reesei*, *Cordyceps militaris*, *Aspergillus oryzae*, and *Claviceps purpurea* [20,21].

The plasmid-dependent CRISPR/Cas9 systems of *G. lucidum* have low versatility between strains due to the different codon biases of genomes [5,10,11,12]. The reported RNP-mediated CRISPR/Cas9 system of *G. lucidum* was constructed for the wild strain and had a relatively low editing efficiency [13]. Moreover, it is uncertain whether this genome editing system could be applied in *G. lucidum* ‘Hunong No. 1’, which is widely cultivated in China. This study constructed an RNP-mediated CRISPR/Cas9 genome editing system in the monokaryotic strain L1 from ‘Hunong No. 1’. This CRISPR/Cas9 system has the advantage of not introducing exogenous genes and exhibits higher editing efficiency compared to the reported CRISPR/Cas9 systems of *G. lucidum* [5,10,11,12,13]. Using this system, two *G. lucidum* functional genes have been successfully edited. This study is of great value to the study of the functional genes of *G. lucidum*.

## 2. Materials and Methods

### 2.1. Strains and Preparation of Protoplasts

The monokaryotic strain L1 from *G. lucidum* ‘Hunong No. 1’ was provided by the Institute of Edible Fungi, Shanghai Academy of Agricultural Sciences. Strain L1 was activated on a potato dextrose agar (PDA) medium (39 g/L PDA powder, BD, Franklin Lakes, NJ, USA) and grown at 26 °C for 7 days. The newly grown mycelia were inoculated into 250 mL flasks containing 100 mL yeast malt glucose (YMG) medium (4 g/L yeast extract, Oxoid, Basingstoke, UK; 10 g/L malt extract, Oxoid, Basingstoke, UK; and 4 g/L glucose, SCR, Shanghai, China) and cultured for 7 days at 26 °C and 150 rpm on a ZWY-2102 rotary shaker (Zhicheng, Shanghai, China). The protoplasts were prepared with reference to the method of Yu et al. and the steps are listed below [22]: mycelia were collected and then washed using sterile water and 109.3 g/L mannitol (SCR, Shanghai, China), followed by digestion with 2% (*w*/*v*) lywallzyme (Guangdong Institute of Microbiology, Guangzhou, China) at 30 °C for 3 h. Protoplasts were collected by centrifugation at 1258× *g* for 6 min. The protoplasts were re-suspended in STC solution (0.55 mol/L sorbitol, Shaoxin Biotech, Shanghai, China; 10 mmol/L CaCl_2_, Sigma, St. Louis, MO, USA; 10 mmol/L Tris-HCl, Leagene Biotech, Beijing, China, pH 7.5) and diluted to a concentration of 10^7^ protoplasts per 100 μL.

### 2.2. The 5-FOA Lethal Experiment on G. lucidum Strain L1

To study the lethal concentration of 5-FOA on the mycelia and protoplasts of *G. lucidum* strain L1, the mycelia and protoplasts of L1 were separately cultivated on PDA media containing 0, 400, 500, 600, 700, 800, 1000, and 1200 mg/L 5-FOA (Sangon Biotech, Shanghai, China). To investigate whether the “lethality” of the mycelia and protoplasts is caused by the solvent dimethyl sulfoxide (DMSO) of 5-FOA, the mycelia and protoplasts of L1 were separately cultured on PDA media containing 0, 0.4, 0.5, 0.6, 0.7, 0.8, 1.0, and 1.2% DMSO (Shaoxin Biotech, Shanghai, China). The content of DMSO corresponds to the concentration of 5-FOA one by one. To maintain the stability of the protoplasts, 0.6 mol/L mannitol was used as an osmotic stabilizer during cultivation of the protoplasts.

### 2.3. Sequencing of ura3, cyp512a3, and cyp5359n1 from G. lucidum Strain L1

The genomic DNA of *G. lucidum* strain L1 was extracted using a DNA extraction kit (Magen, Guangzhou, China). The primers *ura3*-F/*ura3*-R, *cyp512a3*-F/*cyp512a3*-R, and *cyp5359n1*-F/*cyp5359n1*-R (Appendix A) were used to amplify the sequences containing the full length of *ura3*, *cyp512a3*, and *cyp5359n1*. The amplified products were subjected to Sanger sequencing (Genewiz, Suzhou, China).

### 2.4. Preparation of sgRNA-ura3 and Cas9 Protein

The sgRNA-*ura3* was transcribed in vitro using 589 bp downstream of the *ura3* start codon as the target sequence [5]: the sgRNA transcription cassette (T7 promoter-Spacer-sgRNA scaffold) was synthesized by Genewiz, Inc. (Suzhou, China) (Appendix A), and transcribed in vitro by the HiScribe T7 High Yield RNA Synthesis Kit (NEB, Beverly, MA, USA). The Cas9 protein tagged with a nuclear localization signal was purchased from Novoprotein, Inc. (Shanghai, China).

### 2.5. In Vitro Cas9 Cleavage Assay

An in vitro cleavage assay was performed to determine the activity of sgRNA-*ura3*. The primers *ura3*-F/*ura3*-R (Appendix A) were used to amplify the sequence containing the full length of *ura3*. The in vitro cleavage assay was conducted according to the instruction provided by Novoprotein, Inc. (Shanghai, China): 200 ng PCR-amplified fragment, 1 μL Cas9 protein, 200 ng sgRNA-*ura3*, 2 μL of 10 × reaction buffer, and nuclease-free water were added in the 20 μL reaction system. The mixture was incubated at 37 °C for 1 h and then at 70 °C for 10 min. Gel electrophoresis was performed using an appropriate amount of loading buffer (Yeasen Biotech, Shanghai, China) to analyze the results.

### 2.6. PEG-Mediated Transformation of Protoplasts

With a molar ratio of 1:3 between the Cas9 protein and sgRNA, the RNP complexes were assembled on ice in a 20 μL reaction system containing 2–8 μg Cas9 protein (12.5–50 pmol), 1.2–4.8 μg sgRNA (37.5–150 pmol), 2 μL 10 × reaction buffer and nuclease-free water. The mixture was incubated at 37 °C for 15 min to allow RNP complex formation. For PEG-mediated transformation, the following steps were performed referring to the methods described by Yu et al. and Zhang et al. [22,23]: 100 μL STC containing 10^7^ protoplasts, 20 μL RNP complexes, and 50 μL PTC (600 g/L PEG 4000, Sigma, St. Louis, MO, USA; 10 mmol/L Tris-HCl pH 7.5; 50 mmol/L CaCl_2_) were mixed with 0.006% (final concentration) of Triton X-100 (Biofroxx, Einhausen, Germany) on ice and incubated for 10 min. An amount of 1 mL PTC was gently added and mixed in the transformation system, then incubated at 20 °C for 30 min. The mixture was added to an MM regeneration medium (20 g/L glucose; 0.5 g/L magnesium sulfate heptahydrate, SCR, Shanghai, China; 0.46 g/L potassium dihydrogen phosphate, SCR, Shanghai, China; 1 g/L dipotassium hydrogen phosphate, SCR, Shanghai, China; 0.125 mg/L vitamin B1, Sangon Biotech, Shanghai, China; 109.3 g/L mannitol; 20 g/L asparagine, Sangon Biotech, Shanghai, China; 100 mg/L uracil, Sangon Biotech, Shanghai, China; and 10 g/L low-melting-point agarose, Shaoxin Biotech, Shanghai, China) and incubated at 26 °C for 48 h. Afterwards, the medium was covered with a selective MM medium containing 400 mg/L 5-FOA and incubated at 26 °C for 10 days.

When editing the *G. lucidum* functional genes *cyp512a3* and *cyp5359n1*, 10 μg donor DNAs were added in the transformation system. Then, 1 mL PTC was added and gently mixed, followed by incubation at 20 °C for 50 min. Subsequently, an MM regeneration medium without uracil was added and cultured at 26 °C. The preparation process of the donor DNAs was as follows: The target sequence for each gene was selected by http://crispor.tefor.net/ (accessed on 30 November 2021). The L1 genomic DNA was used as a template to assemble the donor DNAs by overlapping PCR using 2 × EasyTaq^@^ PCR SuperMix (TransGen, Beijing, China). The donor DNA contained 500 bp flanking sequences on both the 5′ and 3′ sides of the target sequence, as well as the *ura3* expression cassette. The relevant primers are listed in Appendix A.

### 2.7. Screening and Verification of Transformants

The genomic DNAs of the transformants were extracted using a DNA extraction kit (Magen, Guangzhou, China). PCR amplification was performed using primers E-*ura3*-F/E-*ura3*-R, *cyp512a3*-F/*cyp512a3*-R, and *cyp5359n1*-F/*cyp5359n1* (Appendix A). The edited gene was verified using gel electrophoresis and Sanger sequencing (Genewiz, Suzhou, China).

## 3. Results

### 3.1. The 5-FOA Lethal Experiment on G. lucidum Strain L1

The L1 mycelia grew normally on the PDA medium without 5-FOA. It did not grow on the PDA media containing 400, 500, 600, 700, 800, 1000, or 1200 mg/L 5-FOA. However, it was able to grow on the PDA media containing corresponding concentrations (0.4–1.2%) of solvent DMSO, indicating that the lethal concentration of 5-FOA for L1 mycelia was 400 mg/L, and the “lethal” effect was not caused by the DMSO. The L1 protoplasts (10^7^/Petri dish) were able to regenerate on the PDA media containing 0, 400, 500, 600, 700, or 800 mg/L 5-FOA. However, they could not regenerate on the PDA media containing 1000 or 1200 mg/L 5-FOA. Similar to the mycelia, they were able to regenerate on the PDA media containing the corresponding concentrations (0.4–1.2%) of solvent DMSO, indicating that the lethal concentration of 5-FOA for L1 protoplasts was 1000 mg/L, and the “lethal” effect was not caused by DMSO (Figure 1).

### 3.2. In Vitro Cas9 Cleavage Assay

To verify whether the Cas9 protein could cleave the *ura3* target sequence guided by sgRNA-*ura3*, an in vitro cleavage experiment was conducted. A fragment of 1074 bp, including the *ura3* gene full length of 947 bp, was amplified from the L1 genome using the primers *ura3*-F/*ura3*-R (Figure 2A, Appendix A). Theoretically, cleavage of the 1074 bp fragment by the RNP complex would generate 645 bp and 429 bp sequences (Figure 2A, Appendix A). The agarose gel electrophoresis results indicated that the 1074 bp fragment was cleaved by the RNP complex, producing two fragments consistent with the expected 645 bp and 429 bp (Figure 2B), suggesting efficient cleavage of the *ura3* target sequence by the prepared RNP complex.

### 3.3. Optimization of PEG-Mediated Protoplast Transformation of RNPs on G. lucidum Strain L1

In the RNP-based CRISPR/Cas9 genome editing system, the concentration of RNPs was a key factor that affected editing efficiency. To determine the optimal RNP concentration, different amounts of RNPs (0, 12.5, 25.0, 37.5, and 50.0 pmol) were introduced into the protoplasts of strain L1 via PEG-mediated transformation (10^7^ protoplasts). The concentrations of RNPs in the transformation system (170 μL) were 0.0, 73.5, 147.0, 220.6, and 294.0 nM, respectively. After being cultivated on a double-layer MM medium containing 5-FOA for 10 days, the number of colony-forming units (CFUs) was counted. PCR and Sanger sequencing were performed using the primers E-*ura3*-F/E-*ura3*-R. No transformant was obtained in the transformation system without RNPs. As the concentration of RNPs in the transformation system increased, the number of CFUs obtained also increased (Table 1, Appendix A). 

When the concentration of RNPs was 73.5 nM, 39 CFUs were obtained. Ten of them were randomly selected for sequencing, and mutation was observed near the cleavage site in every transformant. Among them, deletions of 1–44 bp were observed in three mutants (Appendix A), with a deletion rate of 30.0%. Insertions of 1–112 bp were observed in five mutants (Appendix A), with an insertion rate of 50%, including one mutant (20% in five mutants) with a large fragment insertion of more than 20 bp. Substitutions were observed in two mutants: a 65 bp sequence containing PAM was replaced by ATTAACACTA (Appendix A), and a T was replaced by a 72 bp sequence (Appendix A). The substitution rate was 20%.

When the concentration of RNPs was 147 nM, 83 CFUs were obtained. Ten transformants were randomly selected among them for sequencing, and mutation was observed near the cleavage site in every transformant. Among these, deletions of 1–56 bp were observed in seven mutants (Appendix A), with a deletion rate of 70%. Insertions of 2–106 bp were observed in three mutants (Appendix A), with an insertion rate of 30%, including two mutants (66.7% in three mutants) with large fragment insertions.

When the concentration of RNPs was 220.6 nM, 100 CFUs were obtained. Among them, 35 transformants were randomly selected for sequencing, and it was found that all of them had mutations near the cleavage site. Among these, deletions of 1–159 bp were observed in 14 mutants (Appendix A), with a deletion rate of 40%. Insertions of 1–280 bp were observed in 17 mutants (Appendix A), with an insertion rate of 48.6%, including 13 mutants (76.5% in 17 mutants) with large fragment insertions. Different levels of replacements were observed in four mutants: a 52 bp sequence was replaced by a 24 bp sequence (Appendix A), a 261 bp sequence was replaced by a 93 bp sequence (Appendix A), a 46 bp sequence containing PAM was replaced by a 382 bp sequence (Appendix A), and CATAC was replaced by a 569 bp sequence (Appendix A). The replacement rate was 11.4%.

When the concentration of RNPs was 294.0 nM, 111 CFUs were obtained. Among them, 35 transformants were randomly selected for sequencing. It was found that all of them had mutations near the cleavage site. Among these, deletions ranging from 1 to 396 bp were observed in 20 mutants (Appendix A), with a deletion rate of 57.1%. Insertions ranging from 1 to 235 bp were observed in 10 mutants (Appendix A), with an insertion rate of 28.6%, including eight mutants (80% in 10 mutants) with large fragment insertions. Additionally, different degrees of substitutions were observed in five mutants: a 106 bp sequence was replaced with CGTAGACGCCTG (Appendix A), a 52 bp sequence containing PAM was replaced with a 54 bp sequence (Appendix A), CATA was replaced with an 85 bp sequence (Appendix A), C was replaced with an 88 bp sequence (Appendix A), and C was replaced with a 121 bp sequence (Appendix A). The substitution rate was 14.3%.

In conclusion, 220.6 nM and 294.0 nM were the optimal concentrations of RNPs, and they both exhibited editing efficiencies greater than 35 mutants/10^7^ protoplasts for *ura3*, surpassing the reported *G. lucidum* CRISPR/Cas9 genome editing system [5,10,11,12,13]. A schematic diagram of RNP-assisted genome editing in *G. lucidum* is shown in Figure 3.

### 3.4. Analysis of the Repair Characteristics of DSB Induced by RNA-Programmed Nuclease Cas9

Based on transformation with different RNP concentrations, 90 *ura3*-edited mutants were obtained. These mutants were used to research the DSB repair characteristics of *G. lucidum*. Due to the lack of donor DNA containing homologous sequences near the cleavage site in the transformation system, the DSBs induced by RNA-programmed nuclease Cas9 were repaired through non-homologous end joining (NHEJ) or microhomology-mediated end joining (MMEJ), resulting in deletions, insertions, and substitutions. Among them, 46 mutants introduced new sequences through insertions or substitutions near the cleavage site. Regardless of the mutants that only introduced 1–2 bp sequences, the sequence origin analysis was conducted on the 35 mutants that introduced 10–569 bp sequences in the L1 genome. It was discovered that the newly introduced sequences of the 22 mutants (62.9% in 35 mutants) were from the L1 genome itself (Appendix A). Comparing the flanking sequences between the source location of the newly introduced sequence and the cleavage site of *ura3*, no expected MMEJ was found. 

In summary, among the 90 edited mutants, 85 mutants (94.4%) repaired DSBs through NHEJ. Five mutants (5.6%), containing homologous sequences of 3–10 bp near the cleavage site, repaired DSBs through MMEJ and generated deletions of 11–396 bp (Figure 4). Table 2 contains the distribution of NHEJ and MMEJ in the transformation systems with different RNP concentrations. For NHEJ-repaired mutants, 11.8%, 11.8%, 37.6%, and 38.8% of mutants were found in the transformation systems containing RNPs of 73.5, 147.0, 220.6, and 294.0 nM, separately. For MMEJ-repaired mutants, 60.0% and 40.0% of mutants were found separately in transformation systems containing RNPs of 220.6 and 294.0 nM, and no mutant was found with RNPs of 73.5 and 147.0 nM. The above results indicated that both types of repair mechanism had a higher proportion in RNPs of 220.6 and 294.0 nM compared with those of 73.5 and 147.0 nM, possibly due to a greater number of mutants being used for analysis in the two transformation systems. 

### 3.5. Editing of Functional Genes in G. lucidum Using the Constructed CRISPR/Cas9 System

Different from RNA interference technology, the CRISPR system can completely knock out a gene and has been used for researching gene function [6,14]. A few *G. lucidum* function genes have been disrupted and studied using a plasmid-dependent CRISPR system containing exogenous genes [10,12]. In this study, *G. lucidum* function genes were firstly edited using an RNP-mediated CRISPR/Cas9 genome editing system. One mutant with disrupted *ura3*, named L1-Δ*ura3*, was randomly selected as the starting strain for the editing of functional genes in *G. lucidum*. The *cyp450* genes *cyp512a3* and *cyp5359n1* showed upregulated expression in high triterpenoid *G. lucidum* materials (unpublished data). This study selected these two genes to verify the feasibility of editing *G. lucidum* functional genes using the RNP-based CRISPR/Cas9 system. Using the primers *cyp512a3*-F/*cyp512a3*-R, a 2418 bp fragment was amplified from the L1 genome, including the full-length *cyp512a3* gene of 2119 bp, which contains 11 exons. Using the primers *cyp5359n1*-F/*cyp5359n1*-R, a 2059 bp fragment was amplified from the L1 genome, including the full-length *cyp5359n1* gene of 1968 bp, which contains 9 exons (Appendix A). According to the http://crispor.tefor.net/ (accessed on 30 November 2021), the target sequence of *cyp512a3* was selected as 159 bp downstream of ATG, located on its second exon. And the target sequence of *cyp5359n1* was selected as 550 bp downstream of ATG, located on its third exon (Appendix A). The sgRNA transcription cassette (T7 promoter-Spacer-sgRNA scaffold) was synthesized separately (Appendix A). sgRNA-*cyp512a3* and sgRNA-*cyp5359n1* were obtained by in vitro transcription and subsequently assembled with the Cas9 protein, resulting in RNP complexes. RNP (220.6 nM) and corresponding donor DNA (10 μg) were co-transformed into L1-Δ*ura3* protoplasts (10^7^) and selected on an MM medium without uracil. RNPs induced DSBs in the target sequence, and the donor DNA repaired the DSBs through homologous recombination (HR) with the flanking sequences on both the 5′ and 3′ sides of the target sequence. The addition of the *ura3* cassette enabled the mutants to grow on the MM medium without uracil, and served as a marker for the selection of mutants (Figure 5). The donor DNA sequences of *cyp512a3* and *cyp5359n1* are listed in Appendix A. Several CFUs grew on the MM medium for editing *cyp512a3* and *cyp5359n1* (Appendix A). After amplification and sequencing, it was found that two mutants repaired DSBs near the target sequence of *cyp512a3*, and added a *ura3* cassette, named L1-Δ*cyp512a3.1* and L1-Δ*cyp512a3.2*. And the other two mutants repaired DSBs near the target sequence of *cyp5359n1*, and added a *ura3* cassette, named L1-Δ*cyp5359n1.1* and L1-Δ*cyp5359n1.2* (Figure 6). The sequences of the four mutants amplified using the primers *cyp512a3*-F/*cyp512a3*-R or *cyp5359n1*-F/*cyp5359n1*-R are listed in Appendix A. These results indicate that this RNP-based CRISPR/Cas9 system could be used for editing functional genes in *G. lucidum*.

## 4. Discussion

*G. lucidum* is a fungus of high genetic diversity, with significant differences between different strains [24]. Therefore, constructing a genome editing system for cultivated strains had greater significance for the *G. lucidum* industry compared to wild strains. *G. lucidum* ‘Hunong No. 1’ has strong anti-bacterial capabilities, high spore production, and high biological transformation efficiency, and accounts for 70% of high spore-producing varieties [11]. This study used the monokaryotic strain L1 from ‘Hunong No. 1’, and constructed an RNP-assisted CRISPR/Cas9 genome editing system. The editing efficiency of *ura3* was greater than 35 mutants/10^7^ protoplasts, which was higher than those of previously reported *G. lucidum* editing systems [5,10,11,12,13]. The purpose of constructing the *G. lucidum* genome editing system was to study the functions of different genes. In this study, the L1 mutant (L1-Δ*ura3*) with the *ura3* gene disrupted was used as the starting strain. Donor DNA containing homologous arms of the target sequence and *ura3* cassette was added in the transformation system. When RNPs caused DSBs in the target sequence, HR occurred between the donor DNA and L1 genome near the target sequence, resulting in the addition of *ura3* into the genome. At this time, the mutant could grow on a uracil-free MM medium as a marker to screen for mutants, in which the target gene has been edited. Two *G. lucidum cyp450* genes have been edited using this system, indicating more *G. lucidum* genes could be researched by this system.

This study found that the lethal concentration of 5-FOA for L1 mycelia was 400 mg/L, and for L1 protoplasts was 1000 mg/L. The possible reason was that the osmotic pressure in the protoplasts medium containing mannitol was higher than in the mycelia medium, resulting in more 5-FOA being needed to penetrate the medium and enter the protoplasts. In the pre-experiment of RNP transformation, when we added the transformation system to the MM regeneration medium containing 400 mg/L 5-FOA, hundreds of CFUs appeared after several days. Randomly selected CFUs were sequenced, but no edited mutant was found, indicating that the concentration of 400 mg/L was too low to select edited mutants. When we added the transformation system to the MM regeneration medium containing 1000 mg/L 5-FOA, no CFUs appeared after several days, indicating that the concentration of 1000 mg/L was too high and inhibited the regeneration of all protoplasts. Selecting an appropriate concentration of 5-FOA was crucial for the success of the RNP transformation experiment. Finally, the double-layered MM medium was used, with a lower medium without 5-FOA for protoplast regeneration and an upper medium containing 400 mg/L 5-FOA for the selection of edited mutants. The positive transformants would penetrate the upper layer and grow. The editing efficiency of the positive transformant in this study was 100%, indicating that the double-layered medium selection method efficiently screened out edited mutants.

Triton X-100 is a surfactant that could alter the pore size and enhance the permeability of cell membranes [25,26]. Zou et al. first introduced it into the CRISPR/Cas9 transformation system of filamentous fungi [20]. By the addition of 0.006% Triton X-100 (*w*/*v*), the editing efficiency of the *T. reesei ura5* gene was increased from 6 to more than 12 mutants in 2 × 10^5^ protoplasts. Similarly, by the addition of 0.006% Triton X-100 (*w*/*v*), the editing efficiency of the *G. lucidum ura3* gene was increased from 4 to more than 18 mutants in 10^7^ protoplasts [11]. In this study, 0.006% Triton X-100 (*w*/*v*) was chosen to be added into the transformation system, achieving an editing efficiency of greater than 35 mutants/10^7^ protoplasts for the *ura3* gene. Different from that, Eom et al. added 0.95% (*w*/*v*) of Triton X-100 in the transformation system of *G. lucidum* [13], which was more than 150 times the concentration of 0.006%. Overloading Triton X-100 could cause complete disruption of the cell membrane and result in cell death, which might be the main reason for their low editing efficiency (0–22 mutants/10^7^ protoplasts). The 0.006% concentration of Triton X-100 was selected and demonstrated to achieve a high editing efficiency in this study. It indicated that at this concentration, Triton X-100 effectively increased the pore size of *G. lucidum* protoplasts, facilitating the passage of RNPs across the cell membrane and nuclear membrane without causing excessive protoplast death.

Some traits of organisms are controlled by multiple minor-effect genes, such as yield, quality, and disease resistance. The cumulative effects of multiple minor-effect genes can alter the traits of organisms [27,28]. To achieve clear phenotypic changes in these quantitative traits, it is necessary to simultaneously edit multiple associated genes. The CRISPR system, which can simultaneously edit multiple genes, is of great significance for studying the quantitative traits of *G. lucidum*. By modifying the CRISPR system constructed in this study, it is possible to achieve multi-gene continuous editing in *G. lucidum*. The donor DNA is modified to “5′ flank-*ura3* cassette-loop out arm-3′ flank”. The loop out arm has the same sequence as the upstream sequence of the 5′ flank. It is used for “looping out” a *ura3* cassette under the pressure of 5-FOA after the target gene disruption. So far, the marking retrieval is complete, and a new round of gene editing could be conducted. Eleven genes in *T. reesei* were consecutively disrupted using the above approach [29].

## 5. Conclusions

In this study, an RNP-mediated CRISPR/Cas9 genome editing system for a *G. lucidum* cultivated strain was developed and optimized. The editing efficiency of *ura3* was greater than 35 mutants/10^7^ protoplasts, surpassing the previously reported *G. lucidum* CRISPR systems [5,10,11,12,13]. This study revealed the repair characteristics of DSB induced by RNA-programmed nuclease Cas9. The majority was repaired by NHEJ, while a minority was repaired by MMEJ. Two *G. lucidum* genes *cyp512a3* and *cyp5359n1* have been edited using this system. Further efforts should be made toward gene function research using the CRISPR system.

## Figures and Tables

**Figure 1 jof-09-01170-f001:**
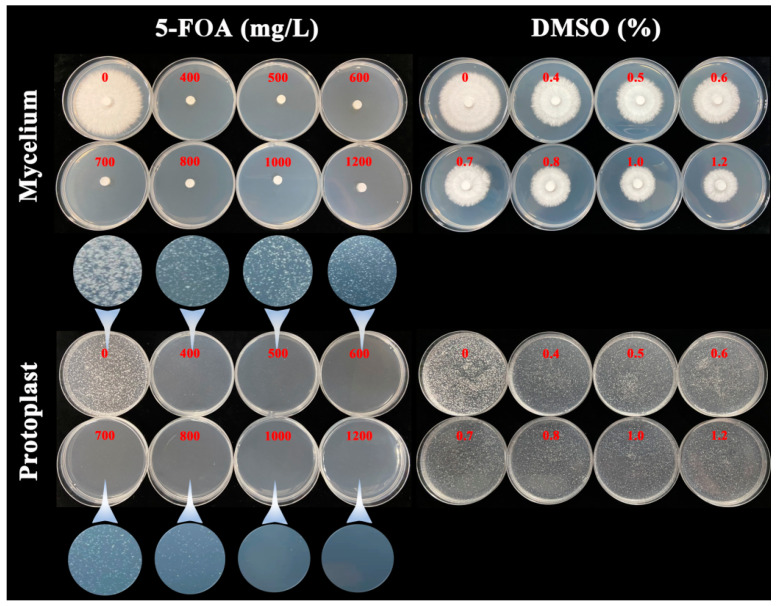
Lethal experiment of 5-fluoroorotic acid (5-FOA) and dimethyl sulfoxide (DMSO) on *Ganoderma lucidum* L1 mycelia and protoplasts. For the four columns of potato dextrose agar (PDA) media on the left, the red numbers represent the concentrations of 5-FOA contained: 0, 400, 500, 600, 700, 800, 1000, and 1200 mg/L. For the four columns of PDA media on the right, the red numbers represent the concentrations of DMSO contained: 0%, 0.4%, 0.5%, 0.6%, 0.7%, 0.8%, 1.0%, and 1.2%.

**Figure 2 jof-09-01170-f002:**
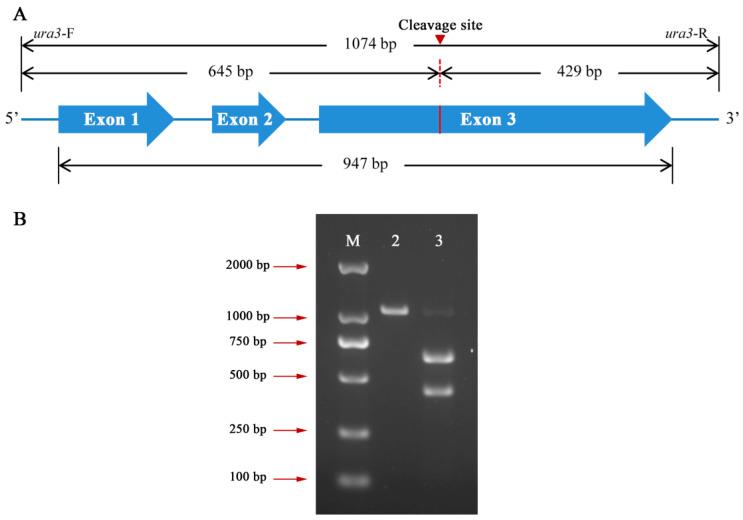
Schematic diagram of the orotidine 5′-monophosphate decarboxylase gene (*ura3*) and in vitro Cas9 cleavage assay. (**A**) The 1074 bp fragment, containing a *ura3* full length of 947 bp, was amplified by primer *ura3*-F/*ura3*-R. The red triangle indicates the cleavage site. The blue arrows represent the exons of the *ura3* gene (**B**) Agarose gel electrophoresis of in vitro cleavage assay. M: DS 2000 maker; 2: *ura3*-1074 bp without ribonucleoproteins (RNPs); 3: *ura3*-1074 bp with RNPs.

**Figure 3 jof-09-01170-f003:**
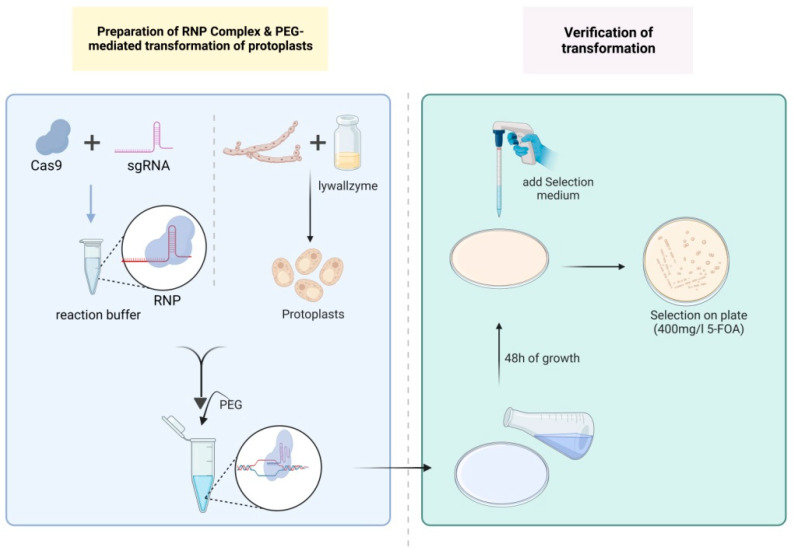
Schematic diagram of RNP-assisted genome editing in *G. lucidum*. (Created By Biorender: Science Suite Inc., Toronto, ON, Canada). The Cas9 protein and single-guide RNA (sgRNA) were assembled in reaction buffer to form RNP. The *G. lucidum* mycelia were treated with lywallzyme and protoplasts were generated. RNP complexes were added to 100 μL protoplast suspensions (containing 10^7^ protoplasts). After addition of Polyethylene Glycol 4000 (PEG 4000) solution, the surfactant Triton X-100 was included at a final concentration of 0.006% (*w*/*v*). The mixture was added in MM regeneration medium and incubated for 48 h. Then the medium was covered with selective MM medium containing 400 mg/L 5-FOA and incubated for 10 days.

**Figure 4 jof-09-01170-f004:**
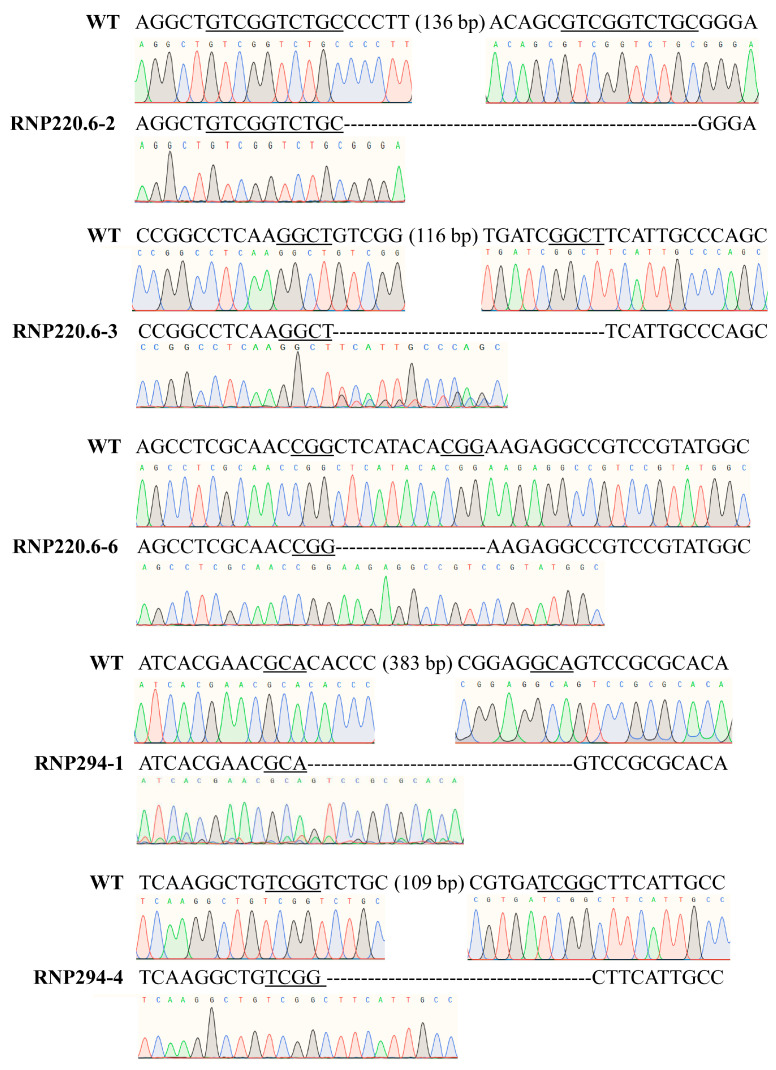
Deletions caused by MMEJ repair. Sequences and corresponding chromatograms of *ura3* in wild-type (WT) and mutants are listed. Underlines indicate sequences with microhomology.

**Figure 5 jof-09-01170-f005:**
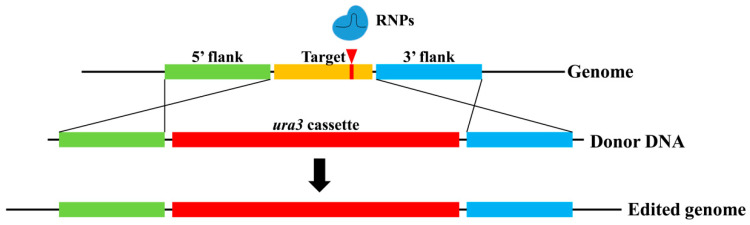
Schematic diagram of functional gene deletion using the RNP-assisted genome editing system in *G. lucidum*. Mutant L1-Δ*ura3* with the *ura3* gene disrupted was used as the starting strain. When RNPs caused DNA double-strand breaks (DSBs) in the target sequence, homologous recombination (HR) occurred between the donor DNA and L1 genome near the target sequence. And the target sequence was replaced by the *ura3* cassette.

**Figure 6 jof-09-01170-f006:**
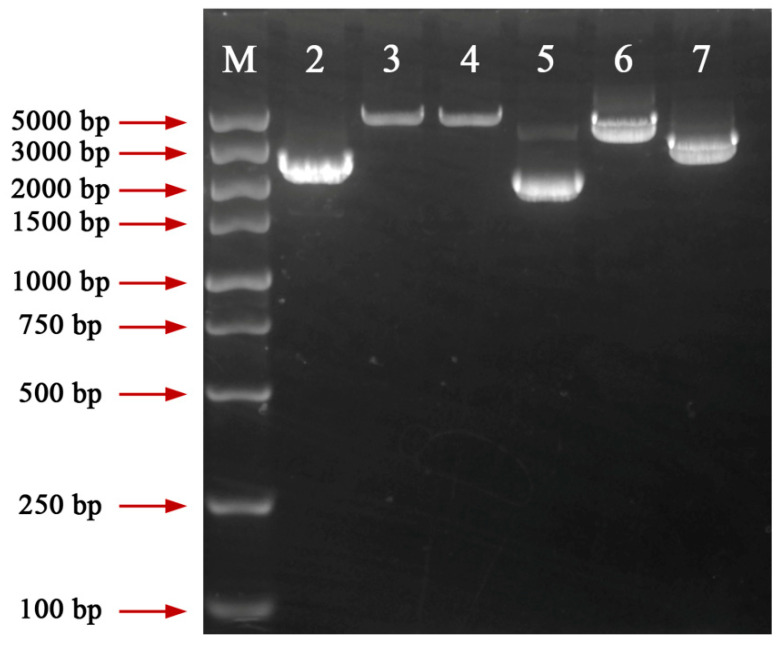
Agarose gel electrophoresis of *cyp512a3* and *cyp5359n1* transformants obtained by co-transformation of donor DNA and RNPs. M: DS 5000 maker; 2: Fragment amplified in L1 using *cyp512a3*-F/R; 3: Fragment amplified in L1-Δ*cyp512a3*.1 using *cyp512a3*-F/R; 4: Fragment amplified in L1-Δ*cyp512a3*.2 using *cyp512a3*-F/R; 5: Fragment amplified in L1 using *cyp5359n1*-F/R; 6: Fragment amplified in L1-Δ*cyp5359n1.1* using *cyp5359n1*-F/R; 7: Fragment amplified in L1-Δ*cyp5359n1.2* using *cyp5359n1*-F/R.

**Table 1 jof-09-01170-t001:** CRISPR/Cas9-based mutagenesis on *ura3* using different concentrations of RNPs.

RNP (nM)	Colony-Forming Units (CFUs)	Deletion Rate	Insertion Rate	Proportion of Large Fragments among All Insertions	Substitution Rate
0.0	0	0	0	0	0
73.5	39 (10/10) ^a^	30.0% (3/10) ^b^	50.0% (5/10) ^c^	20.0% (1/5) ^d^	20.0% (2/10) ^e^
147.0	83 (10/10) ^a^	70.0% (7/10) ^b^	30.0% (3/10) ^c^	66.7% (2/3) ^d^	0
220.6	100 (35/35) ^a^	40.0% (14/35) ^b^	48.6% (17/35) ^c^	76.5% (13/17) ^d^	11.4% (4/35) ^e^
294.0	111 (35/35) ^a^	57.1% (20/35) ^b^	28.6% (10/35) ^c^	80.0% (8/10) ^d^	14.3% (5/35) ^e^
Total	333 (90/90) ^a^	48.9% (44/90) ^b^	38.9% (35/90) ^c^	68.6% (24/35) ^d^	12.2% (11/90) ^e^

^a^ Number of mutants edited/number of transformants sequenced. ^b^ Number of mutants with deletion/number of mutants edited. ^c^ Number of mutants with insertion/number of mutants edited. ^d^ Number of mutants with insertion more than 20 bp/number of mutants with insertion. ^e^ Number of mutants with substitution/number of mutants edited.

**Table 2 jof-09-01170-t002:** Two repair mechanisms distributed in transformation systems containing different RNP concentrations.

Repair Mechanism	RNP (nM)			
	73.5	147	220.6	294
NHEJ	11.8% (10/85) ^a^	11.8% (10/85) ^a^	37.6% (32/85) ^a^	38.8% (33/85) ^a^
MMEJ			60.0% (3/5) ^b^	40.0% (2/5) ^b^

^a^ Number of mutants repaired by non-homologous end joining (NHEJ)/total number of mutants repaired by NHEJ. ^b^ Number of mutants repaired by microhomology-mediated end joining (MMEJ)/total number of mutants repaired by MMEJ.

## Data Availability

Data are contained within the article.

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
