# Peer review of "An Efficient CRISPR/Cas9 Genome Editing System for a Ganoderma lucidum Cultivated Strain by Ribonucleoprotein Method"

_jof, 2023, doi:10.3390/jof9121170_

Round 1

Reviewer 1 Report

Comments and Suggestions for Authors

Tan and co-authors have optimized the transformation and genome editing procedure for the mushroom-forming fungus Ganoderma lucidum. Although this method has been previously published for another strain, the authors have found that the differences between the strains were large enough to require an altered protocol. To develop the protocol, the gene ura3 was deleted. Next, two additional genes were deleted, although no phenotyping was performed on these genes. The paper is therefore purely a technical paper, describing an adjusted methodology.

In principle the experiments seem to have been performed correctly. As such, I only have minor points.

Line 95. “The plasmid-dependent CRISPR/Cas9 systems of G. lucidum [5,10-12] has low universality between strains due to different codon preferences of genomes.” I strongly doubt that the strains are so different from each other that they have a significant difference in codon preference. It seems more likely that this is a result of differences in gene prediction approaches between the two annotated assemblies.

Line 93. “It has been reported in filamentous fungi, such as Trichoderma reesei and Cordyceps militaris [19]”. The first mushroom-forming fungus for which RNPs were developed was S. commune, which may be worth mentioning here.

Line 215. “Lucidum” should be written with a lower case L.

Line 335 (and further). I’m not sure what the authors mean with “the backfill of ura3”. I assume that the gene ura3 was crossed back? Or complemented?

Reviewer 2 Report

Comments and Suggestions for Authors

Considering the unshakable position of Ganoderma in traditional Chinese medicine and the difficulty of implementing gene editing in Mushroom, this manuscript presents a fascinating research work. Tan et al. reported an improved RNP-based CRISPR/Cas9 in Ganoderma lucidum cultivated strain. With the highest efficiency currently reported, two P450 genes cyp512a3 and cyp5359n1 were deleted using this system. And there is no need to rely on knockout Ku70/80 to eliminate the NHEJ pathway. This study provided an important tool kit for genetic breeding of G. lucidum. However, it still needs to be revised before it can be accepted.

1.     Line 18: L1 is not a cultivated strain but Hunong no.1 is.

2.     Line 99-100: It has been reported in filamentous fungi, such as Trichoderma reesei, Cordyceps militaris and Flammulina filiformis [19-20]. Aspergillus oryzae and Claviceps purpurea (PMID: 35224234) also used the similar method.  Please add them.

3.     L102: “universality” should be “versatility”

4.     L103: “preferences” should be “bias”

5.     L106: Delete “Based on the above situation”

6.     L116 “nstitute of Edible Fungi of Shanghai Academy of Agricultural Sciences” should be “Institute of Edible Fungi, Shanghai Academy of Agricultural Sciences”

7.     L122 “mycelia was”should be “mycelia were”

8.     L125 “1258 g” should be “1258 × g

9.     Figure 1: All figures need detailed legends including elucidated “arrows”, “numbers”, and others.

10.  Describe the operation procedures step by step in the legend of Figure 3

11.  Prepare a colored Figure 4 with chromatogram from DNA sequencers。

12.  Figure 5 and 6 are missing.

13.  The conclusion section is missing, please provide a conclusion on the content of the manuscript.

14.  The reference format is not standardized, and the DOI numbers are not necessary.

Comments on the Quality of English Language

It is OK.
